# Studying a Repair Method of LY12 Aluminum Alloy Plate

**Cheng Lv [1,2], Fenghui Wang [1], Sen Yang [1] and Xiang Zhao [1,***

1    Bio-Inspired and Advanced Energy Research Center, Department of Engineering Mechanics, Northwestern Polytechnical University, Xi'an 710129, China; lvcheng623@163.com (C.L.); fhwang@nwpu.edu.cn (F.W.); yangs19@mail.nwpu.edu.cn (S.Y.)
2    Aircraft Strength Research Institute of China, Xi'an 710065, China
*    Correspondence: xzhao@nwpu.edu.cn

**Abstract:** The difference in the strengthening effect of different thickness reinforcement plates on an LY12 aluminum alloy central defect plate and the load distribution between the reinforcement plate and central defect plate in the reinforcement plate structure were studied. Through a method of equivalent transverse load in a small interval [a + Δa], the equivalent cyclic load of the central defect plate in the reinforced plate structure under different crack lengths was calculated, and then the distribution of the internal load of the reinforced plate structure with a different thickness with the crack propagation was solved. The secondary development program of the finite element software Abaqus was written in Python language, so that Abaqus software can solve the problem of high cycle fatigue. The finite element simulation of the different thickness-reinforced plate structure is carried out by this program. Through the output data, the equivalent cyclic load of the central defect plate in the reinforced plate structure under different crack lengths is calculated. Through three different fitting methods, the mathematical relationship between the equivalent cyclic load Δσ and the crack length a at both ends of the central defect plate in the reinforced plate structure is described. Based on the mathematical relationship and the finite element output data, the fatigue crack propagation life of the reinforced plate structures with different thicknesses is calculated. It is found that, under the same crack conditions, with the increase in the thickness of the reinforced plate, the bearing load of the cracked plate decreases and the life of the cracked plate increases. With the expansion of the crack, the bearing load ratio of the reinforced plate increases. The simulation method is compared with the experimental results to verify its effectiveness.

**Keywords:** crack propagation; damage repair; extended finite element simulation

## 1. Introduction

With the development of the aviation industry, the maintenance and repair of aircraft is more important, especially for civil aircraft, wherein greater attention must be paid to safety and cost saving. When the structure presents small fatigue cracks or fatigue damage, reasonable maintenance and strengthening measures can be taken to prolong the safe service life of the aircraft, save maintenance costs to a large extent, and improve flight safety. For the damage of civil aircraft skin, the main repair method is to rivet the reinforcing plate at the damage site.

At present, the metal surface repair method mainly adopts two schemes, one of which is to use carbon-fiber-reinforced polymer (CFRP) to repair the surface damage of the metal materials. In 2012, Xiao Zhi-Gang et al. used carbon-fiber-reinforced polymer (CFRP) plates to repair the cracked beam joints made of thin-walled rectangular hollow sections (RHSs) [1]. In 2014, Yu Qian-Qian et al. studied carbon-fiber-reinforced polymer (CFRP) materials to repair steel plates at different crack propagation stages [2]. In 2016, Reddy et al. studied the fatigue life and stress changes of steel plates with damage under the combined repair of crack arrest holes and carbon-fiber-reinforced polymer (CFRP) coatings. Through finite element simulation analysis and experimental research, it is

found that, under fatigue load, the stiffness can be improved by selecting the parameters of carbon-fiber-reinforced materials, and then, the stress value of the test piece can be reduced to delay the re-initiation of cracks [3]. Wu Xizhi et al. first used the bond force theory to establish the finite element model of cracked steel plate strengthened with CFRP, and studied the fatigue life and strengthening parameters of the cracked steel plate strengthened with carbon-fiber-reinforced polymer (CFRP). This method can reduce the stress intensity factor at the crack tip and effectively improve the fatigue life of the cracked steel plate [4]. From 2017 to 2018, Liu Jie et al. proposed the application of the thread method and mechanical grinding method to CFRP patches to repair cracked aluminum alloy tubes. The fatigue life, residual stiffness and cyclic creep of the repaired specimens were tested. A reinforcement method using externally bonded fiber-reinforced polymer (FRP) angles was also studied to alleviate the longitudinal fatigue cracking of orthotropic steel bridge deck rib-deck joints [5,6]. From 2020 to 2022, Jie Zhiyu et al. studied the enhancement effect of carbon-fiber-reinforced polymer (CFRP) on the fatigue performance of cracked cruciform welded joints through a numerical analysis and fatigue test. The thermal elastic-plastic finite element model of the cruciform steel welded joint was established by ABAQUS software, and the influence of welding residual stress on fatigue crack propagation was studied [7,8]. In 2021, Mohabeddine proposed an analytical model for the mode I fatigue crack propagation of carbon-fiber-reinforced polymer (CFRP) to repair the centrally fractured tensile (CCT) steel specimens [9]. In 2022, Hou Wenyu et al. conducted tests and finite element analysis on the bending behavior of damaged steel beams strengthened with carbon-fiber-reinforced polymer (CFRP) sheets. It has been verified that the CFRP sheet can be used to repair damaged steel beams [10]. Mayur et al. used CFRP patches for the asymmetric repair of aluminum alloy pre-cracked sheets, and studied the fatigue failure cycle at high temperature and room temperature through simulation and experimentation to verify the effectiveness of repair [11].

Another repair scheme is to apply the same metal material to repair the surface damage of metal materials. In 2006, Armentani used the boundary element and finite element programs to simulate the performance of riveted patch repair applied to cracked panels [12]. In 2014, Alemdar studied the deformation fatigue factors of the beam-cross frame of the bridge through simulation, and evaluated the effectiveness of the newly proposed cost-effective retrofit measures. Through parametric study, the best configuration to prevent the fatigue crack propagation of different lengths in the web gap area was determined [13]. In 2016, Guo Tong et al. found that the longitudinal diaphragm tube buckle plate connection had premature fatigue cracking in the long-span cable-stayed bridge. Through field testing and finite element simulation, the fatigue load of the joint was obtained. Through field testing and finite element simulation, it was found that the bolt channel can significantly improve the fatigue life [14]. In 2018, Akshay applied the extended finite element method (XFEM) to the fatigue and fracture analysis of cracked aluminum plates repaired with different shapes of single boron/epoxy resin. In 2018, Akshay applied the extended finite element method (XFEM) to fatigue and fracture analysis of cracked aluminum plates repaired with different shapes of single boron/epoxy resin. Accurately calculate the stress intensity factor (SIF) of repaired cracked panels with various shape patches [15]. In 2021, Ji Chunming used different combinations of fatigue, impact and repair damage to simulate the actual service conditions of aircraft, and proposed a life prediction model based on strain distribution and damage accumulation theory [16]. In 2022, Song Zhou studied the fatigue crack propagation behavior of the laser deposition repair of TA15 titanium alloy with the reliability of the laser deposition repair of aviation parts as the research object [17]. Ricarda used a multi-objective optimization method to perform optimal repair design on the compressor blade disk. A finite element simulation model was proposed to analyze the stress and HCF performance in the welding-affected zone [18]. Zhang proposed a new repair method for cruciform column base joints. It was verified by experiments that the new repair method has good bearing capacity and stiffness [19]. Wang simulated and analyzed the front frame of the car and repaired the

middle surface through HyperMesh. Through modal analysis and fatigue life analysis, it was verified that the method met the actual requirements [20]. In 2023, Kang used laser cladding (LC) additive manufacturing technology to repair damaged steel structures. Through finite element simulation and experimental analysis, it was verified that the repair method restored the stiffness, strength and geometric dimensions of the damaged structure to the undamaged state [21]. Qiang proposed a new method of iron-based shape memory alloy (Fe-SMA) plate covering the crack arrest hole to alleviate the stress concentration at the edge of the crack arrest hole. Through simulation analysis and experiment, it was verified that, the thicker the Fe-SMA plate is, the better the repair effect is [22]. This method is used to repair the cracks at the arc incision of the diaphragm of the orthotropic steel deck (OSD) [23].

Because the aircraft, especially the civil aircraft, pays more attention to safety, the selection of maintenance materials is conservative. For a civil aircraft, the maintenance materials mainly select the same materials of the original damaged parts for maintenance. Therefore, this paper mainly studies the repair methods of open cracks that often occur in aircraft skins. The repair performance of the original damaged metal plate is studied by connecting the reinforcing plate with screws. Through the secondary development of Python language, the problem that Abaqus cannot calculate the stress intensity factor in crack propagation is solved. Through the obtained stress intensity factor, the load condition and residual life of the test piece are calculated.

## 2. Basic Theory of Fracture Mechanics and Damage Tolerance

### 2.1. Fatigue Crack Growth Threshold and Fracture Toughness

In fracture mechanics, to determine whether a test specimen begins to experience fatigue failure or whether the crack begins to expand, the usual criterion is to compare the magnitude of the real-time stress intensity factor $\Delta K$ and the fatigue crack propagation threshold $\Delta K_{th}$.

$$\Delta K_{th} = K_{th,0}(1 - R)^{\eta} \tag{1}$$

$$K_{th,0} = \Delta K_{th}(R = 0) \tag{2}$$

In Equations (1) and (2), $\eta$ is the test constant. It can be seen from the formula that the fatigue crack threshold $\Delta K_{th}$ is a variable related to the stress ratio R.

### 2.2. Critical Size of Fatigue Crack

For the fatigue fracture problem in the linear elastic range, the specimen usually has a crack critical value. When the crack critical value is reached, the specimen will break instantaneously. The crack critical value is called the fatigue crack critical size of the specimen, which is usually expressed by the symbol $a_c$. The fatigue crack critical size is usually related to many factors, which can be expressed by Equation (3).

$$a_c = \frac{1}{\pi}\left(\frac{K_c}{f\sigma_{max}}\right) \tag{3}$$

In the equation, $K_c$ is the fracture toughness, the unit is $Mpa\sqrt{m}$, which is related to the material of the specimen, f is the geometric correction coefficient, $\sigma_{max}$ is the maximum load value, and the unit is Mpa.

## 3. Fatigue Damage Tolerance Test and Analysis of LY12 Reinforced Plate Structure

### 3.1. Experimental Design

The material of the test piece used in this paper is the civil aircraft skin material LY12 aluminum alloy. The fatigue fracture performance parameters are shown in Table 1.

**Table 1.** Material properties of LY12 aluminum alloy.

| KIC (MPa$\sqrt{\text{m}}$) | E (GPa) | μ | σs (MPa) | σb (MPa) |
|:---:|:---:|:---:|:---:|:---:|
| 23.2 | 68 | 0.33 | 322 | 443 |

The specimens used in this test are a 4 mm thick central defect plate, 2 mm thick reinforcement plate, a 3 mm thick reinforcement plate, and a 4 mm thick reinforcement plate. As shown in Figure 1, the reinforcement plate is connected to the defect plate by prefabricated bolts.

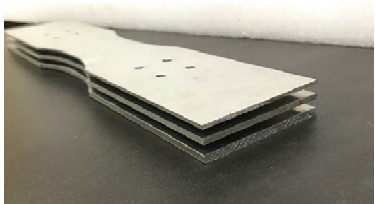

**Figure 1.** The 4 mm thick central defect plate.

### 3.2. Test Methods and Steps

Before the test, a uniform load was applied to the right end of the self-designed 4 mm thick central defect plate by the finite element software Abaqus, and the left end was fixed. The finite element analysis was performed to extract the maximum stress point and test whether the design met the requirements of the fatigue test. The test results are shown in Figure 2.

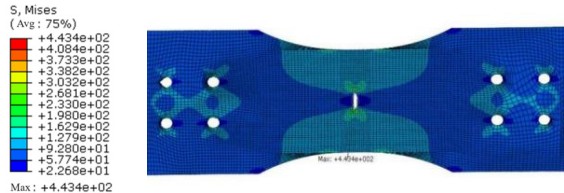

**Figure 2.** Stress cloud diagram for the 4 mm central defect plate.

As shown in Figure 2, the maximum stress point is at the center crack tip, and the crack tip has an obvious plastic zone and stress concentration phenomenon. It can be considered that the crack tip is a dangerous part, and the crack will be generated and expanded here. The bolt hole is not a dangerous part, which will not affect the test results and analysis, meet the test requirements, and prove that this design scheme is feasible.

The fatigue test for the 4 mm thick central defect plate was performed to obtain the test constants C, n related to the material, which provides the basis for subsequent data processing, as shown in Figure 3.

### 3.3. Analysis of Fatigue Damage Tolerance Test Results of LY12 Strengthened Plate Structure

3.3.1. Effect of Different Thickness of Reinforcing Plate on Fatigue Crack Propagation Life

In this test, the fatigue test of the central defect plate with a thickness of 4 mm was first carried out. The fatigue test constants C, n. C, and n are related to the test specimen itself, the test conditions, and the test environment and other factors. The following formula can be derived from the Paris formula:

$$\ln\frac{da}{dN} = \ln C + n\ln\Delta K \tag{4}$$

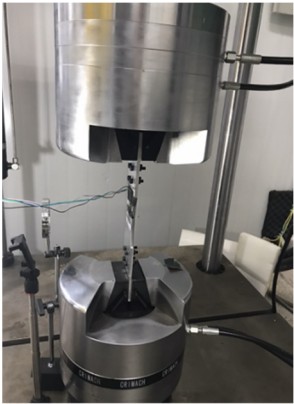

**Figure 3.** Reinforcement plate structure in the test.

It can be seen from Equation (4) that the logarithm of the crack growth rate da/dN of the specimen is linearly related to the logarithm of the stress intensity factor ΔK at the crack tip.

Three tests were carried out on the four test specimens to exclude the occurrence of accidental and random test results. The fatigue crack propagation life is shown in Table 2.

**Table 2.** Fatigue crack propagation life of four test specimens.

| Test Piece Number | 4 mm Defect Plate | 2 mm Reinforced Plate Structure | 3 mm Reinforced Plate Structure | 4 mm Reinforced Plate Structure |
|---|---|---|---|---|
| 1 | 15,510 | 39,166 | 51,573 | 62,694 |
| 2 | 14,297 | 41,350 | 51,315 | 62,751 |
| 3 | 14,564 | 41,280 | 53,121 | 64,650 |
| Mean value | 14,790 | 40,598 | 52,003 | 63,365 |

It can be concluded from Table 2 that the test repeatability is good, so the test data are valid data. With the increase in the thickness of the reinforced plate, the fatigue crack propagation life of the structure will still increase, and the two are positively correlated.

### 3.3.2. Effect of Different Thickness of Reinforcing Plate on Ultimate Crack Length

The fatigue crack limit length ac of the different thickness-reinforced plate structures can be obtained by observation, and the data are recorded in Table 3.

**Table 3.** Four kinds of test specimens fatigue limit crack (unit: mm).

| Test Piece Number | 4 mm Defect Plate | 2 mm Reinforced Plate Structure | 3 mm Reinforced Plate Structure | 4 mm Reinforced Plate Structure |
|---|---|---|---|---|
| 1 | 20 (19.5) | 21 (21.4) | 22.5 (21.7) | 20 (19.5) |
| 2 | 20 (20.3) | 21 (21.2) | 22 (22.2) | 20 (20.3) |
| 3 | 20.1 (20.5) | 21.1 (21.3) | 21.7 (21.9) | 20.1 (20.5) |
| Mean value | 20.07 | 21.17 | 22 | 20.07 |

With the increase in the thickness of the reinforced plate in the reinforced plate structure, the fatigue limit crack length $a_c$ will also increase. The greater the thickness of the strengthened plate, the more obvious the strengthening effect on the central defect plate.

The theoretical formula for calculating the fatigue limit crack length is represented as follows

$$\Delta K_{IC} = \Delta\sigma\sqrt{\pi a_c}g(\vartheta) \tag{5}$$

$$g(\vartheta) = \left(1 - 0.25\vartheta^2 + 0.06\vartheta^4\right)\sqrt{\sec\left(\frac{\pi}{2}\vartheta\right)} \tag{6}$$

$$\vartheta = \frac{2a_c}{W} \tag{7}$$

In Equations (5)–(7), W is the width of the dangerous part of the central defect plate, $g(\vartheta)$ is the geometric correction coefficient, and $a_c$ is the limit crack length.

### 3.3.3. Effect of Different Thickness of Strengthened Plate on Fatigue Crack Growth Rate

It can be seen from Figure 4 that the growth rate of the fatigue crack length of the three reinforced structures increases with the increase in the number of fatigue load cycles, With the increase in the thickness of the reinforced plate, the stiffness of the reinforced plate increases. As a result, the bearing load of the reinforced plate increases and the bearing load of the specimen decreases, so the fatigue performance of the specimen also increases.

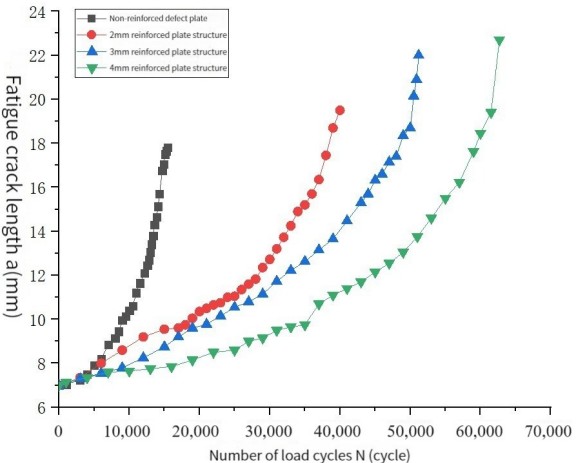

**Figure 4.** Comparison of fatigue crack propagation life curves of different structures.

### 3.3.4. Calculation and Comparison of Stress Intensity Factors of Different Thickness Reinforced Plate Structures

The stress intensity factor is a variable that reflects the size of the stress and strain field at the crack tip of the specimen. As the crack of the specimen expands, the stress intensity factor amplitude $\Delta K$ will also change. $\Delta a$ tends towards 0 or is very small. The equivalent cyclic load at both ends of the central defect plate can be regarded as a constant amplitude load. Then, the Paris formula can be used to calculate the stress intensity factor in the interval $[a, a + \Delta a]$. The stress intensity factor is used to represent the equivalent stress intensity factor when the crack length is a.

The secant method is used to calculate the fatigue crack growth rate. The specific formula is as follows.

$$\frac{da}{dN} = \frac{(a_{i+1} - a_i)}{(N_{i+1} - N_i)} \tag{8}$$

In Equation (8), $a_i$ is the crack length of a point, $a_{i+1}$ is the crack length of adjacent points, $N_i$ is the number of fatigue load cycles corresponding to $a_i$, and $N_{i+1}$ is the number of fatigue load cycles corresponding to the crack length $a_{i+1}$ of adjacent points.

The crack growth rate under each crack length is calculated using Equation (9):

$$\frac{da}{dN} = C(\Delta K)^n \tag{9}$$

The stress intensity factor amplitude $\Delta K$ corresponding to different crack lengths is calculated. In Equation (9), C and n are experimental constants related to the materials, which were obtained through previous experiments. The following diagram can be drawn by calculation.

As shown in Figure 5, as the crack propagates, the stress intensity factor amplitude ΔK of the central defect plate in the reinforced plate structure was calculated according to the test data becoming larger and larger. Under the same crack length, the thicker the thickness of the reinforcing plate is, the larger the bearing load of the reinforcing plate is, the smaller the bearing load of the specimen is, and the smaller the stress intensity factor amplitude ΔK is. This is also one of the reasons why the fatigue performance of the reinforced plate structure increases with the increase in the thickness of the reinforced plate.

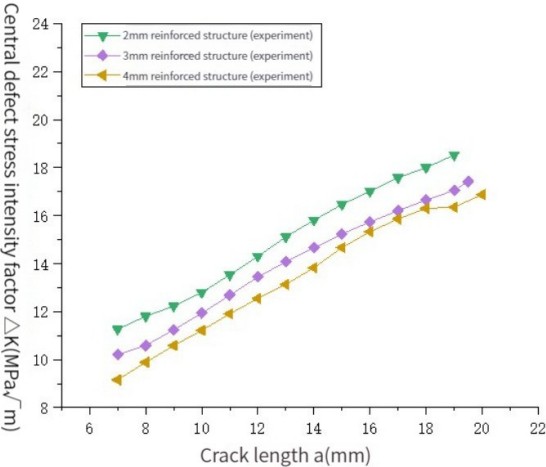

**Figure 5.** The stress intensity factor amplitude ΔK under different crack lengths.

### 3.3.5. Load Analysis of Central Defect Plate with Different Thicknesses and a Reinforced Plate Structure

According to the above results, the uniformly distributed load Δσ at both ends of the central defect plate in the reinforced plate structure can be obtained by Equations (5)–(9) in a small crack change interval. The following figure can be drawn by calculation.

As shown in Figure 6, with the increase in crack length, the equivalent load on both ends of the central defect plate in the reinforced plate structure with different thicknesses is decreasing. Under the same crack length, the larger the thickness of the reinforced plate is, the smaller the equivalent load on the central defect plate is. This also explains why a thicker reinforced plate results in a better the fatigue performance of the structure.

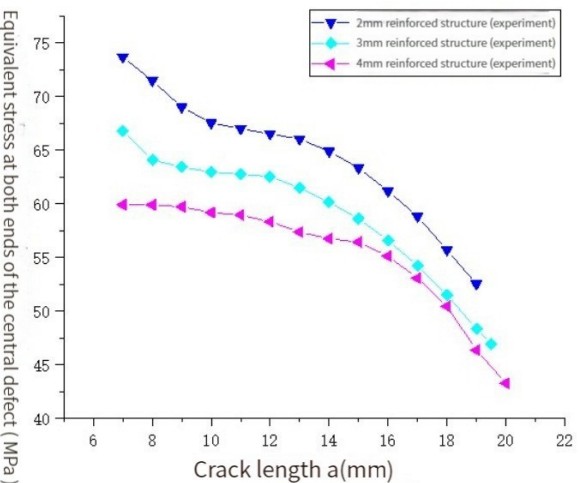

**Figure 6.** The uniform load of equivalent bearing at both ends of the central defect plate in the reinforced plate structure under different crack lengths.

### 3.3.6. Load Analysis of Reinforced Plates with Different Thicknesses in Fatigue Test

By calculating the equivalent load at both ends of the central defect plate in the strengthened plate structure with different thickness under different crack lengths, the equivalent load of the strengthened plate with different thicknesses under different crack lengths can be calculated. With each propagation of the fatigue crack, the strengthened plate and the central defect plate in the strengthened plate structure will realize the redistribution of a load. The load carried by the strengthened plate has been changing with the propagation of the fatigue crack.

It can be seen from Figure 7 that, with the increase in the half-crack length a of the central defect plate in the structure, the load ratio of the reinforcing plate with different thicknesses in the structure is increasing. Under the same crack length, the thicker the reinforcing plate is, the higher the load ratio is, indicating that a thicker reinforcing plate means a more obvious strengthening effect.

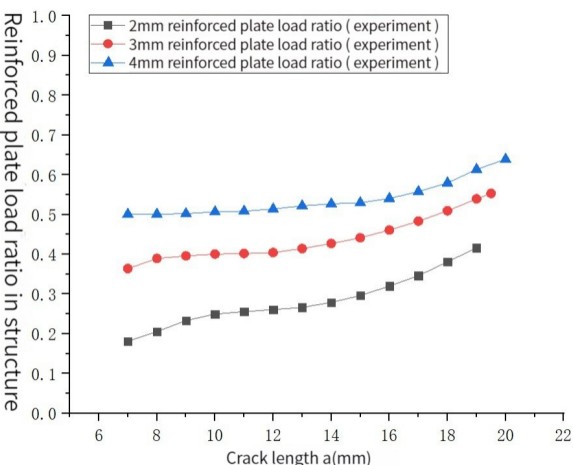

**Figure 7.** The load ratio of different reinforcing plates in the structure.

### 3.3.7. Description of Residual Life of Different Test Specimens

The description of the residual fatigue life of different test pieces can be described by crack length 2a and the width W of the dangerous part of the test piece. When the fatigue crack extends to 2a = W, the test piece breaks. The fatigue crack propagation of the test piece can be described by $1 - (2a/W)$. When $1 - (2a/W) = 0$, the residual fatigue life of the test piece is 0. Because the 4 mm thick central defect plate itself has a central defect in this test, and a crack is preset in advance for the 4 mm thick central defect plate, $1 - (2a/W)$ cannot reach 1, that is, a $\neq$ 0, the fatigue crack residual propagation life = total life of fatigue crack propagation − load cycle number.

As shown in Figure 8, the maximum value of $1 - (2a/W)$ is 0.72 because of the pre-fabrication of 2 mm semi-crack in the central defect plate of 4 mm thickness. Under the same fatigue crack residual propagation life, with the increase in the thickness of the reinforcing plate, the $1 - (2a/W)$ of the reinforcing plate structure decreases, that is to say, with the increase in the thickness of the reinforcing plate, the fatigue resistance of the reinforcing plate structure is improved. From the trend of the curve, the greater the $1 - (2a/W)$ of the same structure, the greater the residual fatigue crack propagation life, that is, the value of $1 - (2a/W)$ can effectively describe the residual fatigue crack propagation life, and the two are positively correlated.

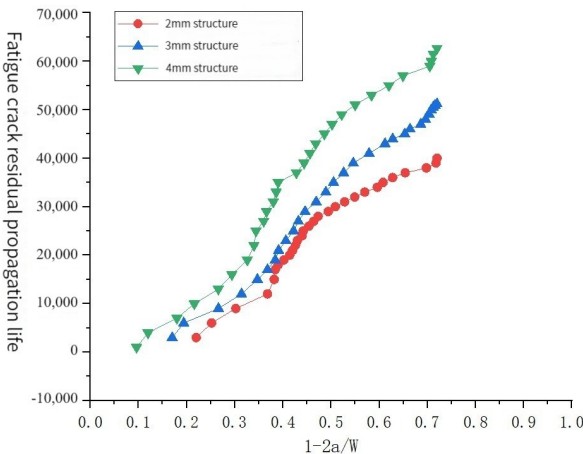

**Figure 8.** Description of residual fatigue crack growth life of stiffened plate structures with different thicknesses.

## 4. Finite Element Analysis of LY12 Reinforced Plate Structure Damage Tolerance Test

### 4.1. Establishment of Finite Element Model

Since the crack tip displacement of the fatigue fracture problem is usually not a continuous problem, the conventional finite element method cannot solve the calculation problem of the fatigue crack tip well, but the extended finite element method can solve the problem of discontinuous crack tip displacement in the fatigue fracture problem. Usually, a special function is added to the discontinuous mode of crack tip displacement, which makes this kind of problem become a continuous problem. In this paper, the extended finite element method is used for simulation analysis.

The defect plate and the reinforcing plate are connected by bolts. In order to ensure that different structures have the same initial far-field stress $\Delta\sigma$, the bearing load of the unreinforced plate is set to 18 kN, the bearing load of the reinforced plate thickness of 2 mm is 18 kN, and the thickness of 3 mm and 4 mm is 21 kN and 2 kN, respectively. The model grid diagram is shown in Figure 9.

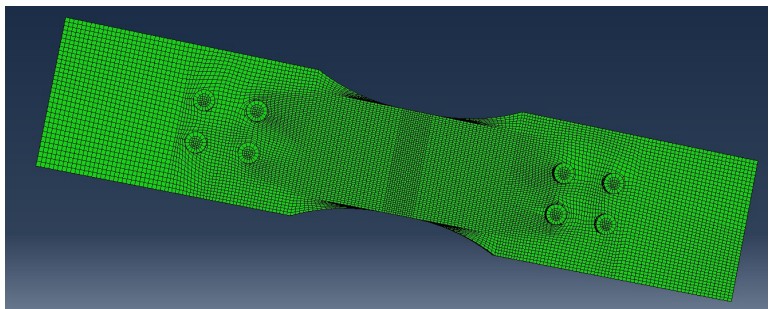

**Figure 9.** Strengthened plate structure model diagram.

### 4.2. Analysis and Data Processing of Finite Element Model

4.2.1. The Stress Intensity Factor Amplitude $\Delta K$ of the Program Output

The secondary development of the Abaqus software is carried out in Python language. Based on the extended finite element principle, the fatigue damage tolerance analysis of the test specimens is automatically carried out. The relationship between the stress intensity factor amplitude $\Delta K$ output by the program and the half-crack length of the central defect plate in the reinforced plate structure is described in the following figure.

As shown in Figure 10, with the increase in the half crack length a of the central defect plate in the reinforced plate structure, the stress intensity factor amplitude $\Delta K$ of the crack tip of the three reinforced plate structures increases. At the same crack length, the greater the thickness of the reinforced plate, the smaller the stress intensity factor amplitude $\Delta K$ of

the crack tip of the central defect plate in the structure, which is similar to the experimental results. The comparison between the stress intensity factor amplitude ΔK at the crack tip calculated by the experimental data and the simulated output ΔK is shown in the following figure.

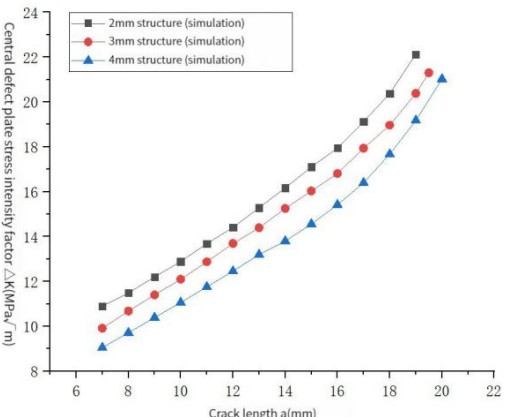

**Figure 10.** The stress intensity factor amplitude ΔK under different crack lengths in the simulation experiment.

As shown in Figures 11–13, by comparing the simulation results of reinforced plate structures with different thicknesses with the experimental data, it can be found that the variation trends in ΔK obtained by the two methods with crack propagation are basically the same, and when the crack length is small, the error between the two is very small. With the increase in crack length, the results of the finite element simulation output are larger than those calculated by experimental data. Through calculation, the average relative errors of ΔK obtained by the two methods are 4.85%, 6.78%, and 4.57%, respectively. In engineering, these fall within the allowable range, indicating that the stress intensity factor amplitude ΔK output by the self-programmed Abaqus secondary development program based on Python language is close to the experimental data.

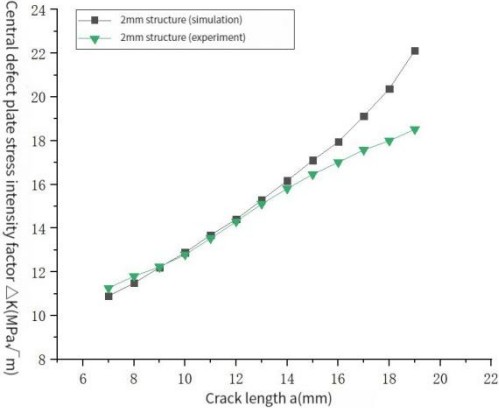

**Figure 11.** Comparison of stress intensity factor amplitude ΔK test and simulation results of 2 mm reinforced plate structure.

### 4.2.2. Limit Crack Length Based on Program Output

Table 4 lists the ultimate crack length of the 2 mm, 3 mm, 4 mm reinforced plate structure based on the final output of the program and compares it with the experimental results.

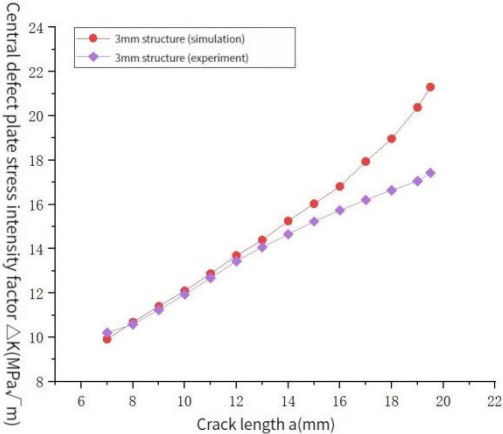

**Figure 12.** Comparison of stress intensity factor amplitude ΔK test and simulation results of 3 mm reinforced plate structure.

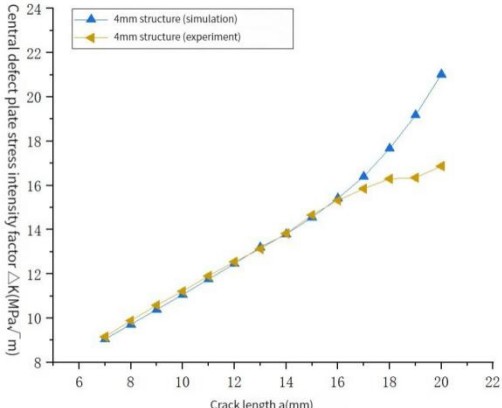

**Figure 13.** Comparison of stress intensity factor amplitude ΔK test and simulation results of 4 mm reinforced plate structure.

**Table 4.** Comparison of limit crack length.

| Reinforced Plate Thickness | Experimental Results | Simulation Results | Relative Error |
| :---: | :---: | :---: | :---: |
| 2 mm | 20.07 | 19 | 5.33% |
| 3 mm | 21.17 | 19.6 | 7.42% |
| 4 mm | 22 | 20.1 | 8.64% |

Through comparison, it can be seen that the limit crack length output by the program is more conservative. The relative error between the limit crack lengths obtained by the three reinforced plate structures in different ways is less than 10%, which meets the engineering requirements.

### 4.2.3. Load Proportion of the Central Defect Plate in Reinforced Plate Structure

According to the program setting, the maximum stress intensity factor $K_{max}$ under each crack length and the stress intensity factor amplitude ΔK under each crack length are obtained, and the equivalent uniform load of the central defect plate at both ends of different reinforced plate structures can be calculated.

As shown in Figure 14, the simulation results are consistent with the experimental results. As the crack length increases, the cross-sectional area of the bearing parts at both ends of the specimen center decreases, resulting in an increase in the equivalent stress. The comparison results are shown in Figures 15–17.

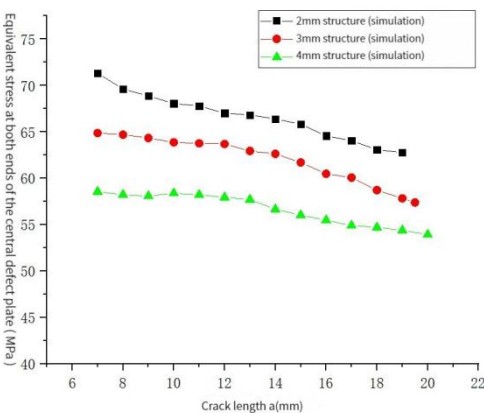

**Figure 14.** Uniform load of equivalent bearing at both ends of central defect plate in the reinforced plate structure under different crack lengths.

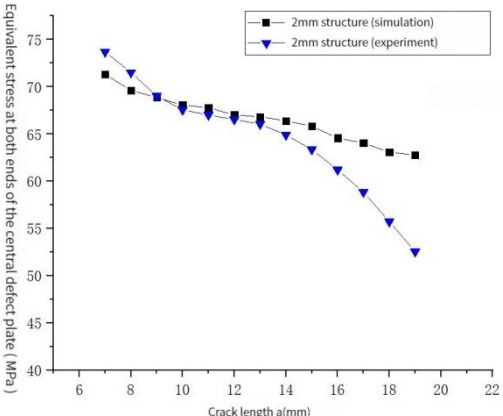

**Figure 15.** Comparison between the central defect plate load distribution test and simulation results for the 2 mm reinforced plate structure.

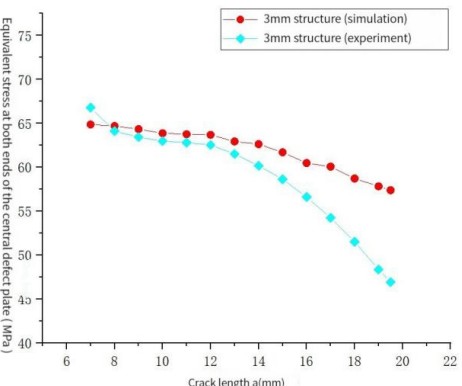

**Figure 16.** Comparison between the central defect plate load distribution test and simulation results for the 3 mm reinforced plate structure.

By comparing the results, it can be found that the finite element simulation results are basically consistent with the experimental results in the stable propagation stage of the crack, and enter the rapid propagation stage. Since the results of the experimental data are based on the Paris formula, the Paris formula can only effectively describe the stable propagation stage of fatigue crack propagation. Therefore, a certain deviation will be reflected in the rapid propagation stage of the crack. However, in the study of damage tolerance, only the stable propagation zone of fatigue crack is usually concerned, and it can be seen that the program output of the simulation experiment is conservative. The average relative errors between the finite element simulation results of different reinforced

plate structures and the experimental data are 4.85%, 6.78%, and 4.57%, respectively. In engineering, this error is within the allowable range, indicating that the simulation method has good practicability.

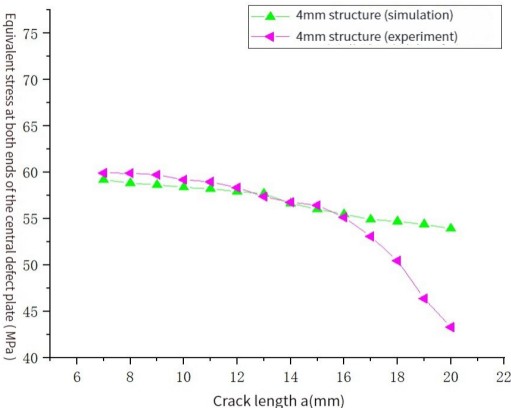

**Figure 17.** Comparison between the central defect plate load distribution test and simulation results for the 4 mm reinforced plate structure.

4.2.4. Mathematical Description of Load Distribution of Plate with Central Defect in Stiffened Plate Structure

Through the previous research, it was found that there is a certain mathematical relationship between the half-crack length a of the central defect plate in the reinforced plate structure with different thicknesses and the load distributed by the central defect plate in the reinforced plate structure. The least square method is used to fit the two variables. Linear fitting, quadratic polynomial fitting, and cubic polynomial fitting are used to describe the mathematical relationship between them, and the functional relationship between the two variables is established. After fitting the polynomial, by calculating the fitting coefficient of determination $R^2$, the closer the value of $R^2$ is to 1, and the closer the fitting result is to the real data.

It can be concluded from Table 5, that for the same reinforced plate structure, the fitting determination coefficient $R^2$ will increase with the increase in the number of fitting polynomials. The higher the number of fitting polynomials, the closer the fitting results will be to the data of finite element simulation. The coefficient of determination is greater than 0.99, indicating that the goodness of fit between the independent variable and the dependent variable is very high and meets the fitting requirements.

**Table 5.** The coefficient of determination $R^2$ of the polynomial is fitted in different ways.

|  | 2 mm Reinforced Plate Structure | 3 mm Reinforced Plate Structure | 4 mm Reinforced Plate Structure |
|---|---|---|---|
| Linear fitting | 0.96773 | 0.93187 | 0.96896 |
| Quadratic polynomial fitting | 0.98798 | 0.98979 | 0.98674 |
| Cubic polynomial fitting | 0.99783 | 0.99606 | 0.99713 |

4.2.5. Prediction and Analysis of Fatigue Crack Propagation Life of Reinforced Plate Structures with Different Thicknesses

When the crack in the central defect plate of the reinforced plate structure expands, as long as the expansion amount Δa is small enough, it can be approximately considered that the equivalent load at both ends of the central defect plate in the reinforced plate structure does not change in a very small interval [a, a + Δa].

Then, using the fatigue crack propagation life accumulation formula:

$$\sum_{i=1}^{L} n_i = N \tag{10}$$

The fatigue life estimation formula based on Paris formula:

$$n_i = \int_{a_i}^{a_i + \triangle a} \frac{da}{C \left( g \Delta \sigma_i \sqrt{\pi a} \right)^n} \tag{11}$$

The estimated value of the fatigue crack's propagation life can be obtained. In Equation (10), L is the number of small intervals, which is the fatigue crack propagation life corresponding to the first interval and the cumulative life of each interval. In Equation (11), C, n are the experimental constants related to the material, g is the geometric correction coefficient, which is the fatigue uniform load amplitude of the corresponding interval, and the unit is Mpa.

The central crack plate of g in Equation (12) for the uniform tensile fatigue test is:

$$g(\vartheta) = \left( 1 - 0.25\vartheta^2 + 0.06\vartheta^4 \right) \sqrt{\sec\left( \frac{\pi}{2}\vartheta \right)} \tag{12}$$

$$\vartheta = \frac{2a}{W} \tag{13}$$

In Equation (13), W is the width of the dangerous part of the central defect plate of the reinforced plate structure with different thicknesses, and the unit is m.

Bring Equations (11)–(13) into Equation (10) to obtain:

$$N = \sum_{i=1}^{L} \frac{1}{C \left( \sqrt{\pi}\Delta\sigma_i \right)^n} \int_{a_i}^{a_i + \triangle a} \frac{da}{\left\{ \sqrt{a} \left[ 1 - 0.25 \left( \frac{2a}{W} \right)^2 + 0.06 \left( \frac{2a}{W} \right)^4 \right] \sqrt{\sec\left( \frac{\pi a}{W} \right)} \right\}^n} \tag{14}$$

Through Equation (14), the fatigue crack propagation life N corresponds to the half crack length of the central defect plate in the reinforced plate structure with different thicknesses, which is obtained when the half crack length of the central defect plate is the limit crack length so as to achieve the purpose of estimating the fatigue crack propagation life.

The fatigue crack growth life is obtained based on the linear relationship, quadratic polynomial, and cubic polynomial, respectively, as shown in Table 6. The errors of the results obtained by different methods relative to the experimental results are shown in Table 7.

**Table 6.** Fatigue crack growth life estimated by different fitting polynomials.

| | 2 mm Reinforced Plate Structure | 3 mm Reinforced Plate Structure | 4 mm Reinforced Plate Structure |
|---|---|---|---|
| Linear fitting | 33,865 | 42,758 | 57,950 |
| Quadratic polynomial fitting | 35,271 | 44,896 | 58,754 |
| Cubic polynomial fitting | 37,041 | 49,281 | 62,552 |
| Experiment average life | 40,598 | 52,003 | 63,365 |

**Table 7.** Relative error of fatigue crack propagation life predicted by different methods.

|  | 2 mm Reinforced Plate Structure | 3 mm Reinforced Plate Structure | 4 mm Reinforced Plate Structure |
|---|---|---|---|
| Linear fitting | 16.59% | 17.78% | 8.55% |
| Quadratic polynomial fitting | 13.12% | 13.67% | 7.28% |
| Cubic polynomial fitting | 8.76% | 5.23% | 1.28% |

By comparing the estimated life of the structure with the average life of the structure, it can be seen that the closest to the experimental data that the fatigue crack propagation life of the reinforced plate structure can be is calculated by the cubic polynomial fitting. It can be seen from Tables 5–7 that the relative error between the fatigue crack propagation life calculated by the cubic polynomial fitting and the real test data are the smallest, and the relative error is less than 9%. The simulation results are conservative and within the allowable error range of engineering.

### 4.2.6. Drawing the Fatigue Crack Growth Life Curve of the Structure of Reinforced Plates with Different Thicknesses

Through the function relationship between the half crack length a and the equivalent cyclic load amplitude Δ at both ends of the central defect plate, which is fitted by linear fitting, quadratic polynomial fitting and cubic polynomial fitting, the number of load cycles corresponding to each crack length of the structures of reinforced plate with different thicknesses can be calculated. The fatigue crack propagation life curve (a-N diagram) of the structures of reinforced plates with different thicknesses is shown in Figures 18–20.

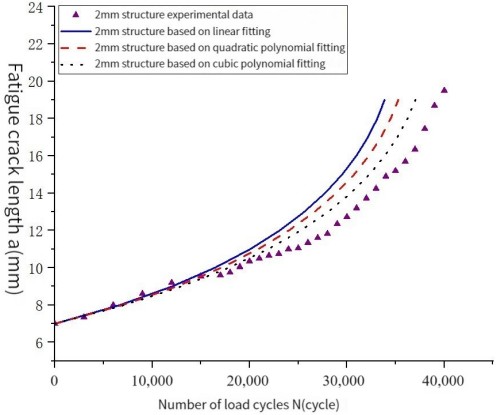

**Figure 18.** Data comparison for the 2 mm reinforced plate structure.

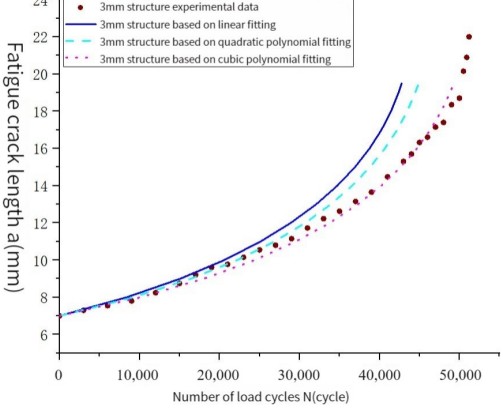

**Figure 19.** Data comparison for the 3 mm reinforced plate structure.

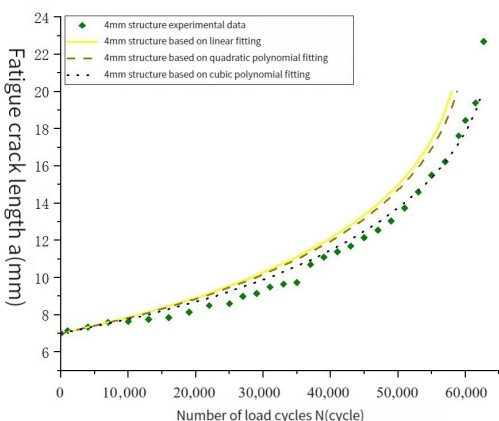

**Figure 20.** Data comparison for the 4 mm reinforced plate structure.

Figures 18–20 show that the fatigue crack propagation life curves of the three reinforced plate structures based on cubic polynomial fitting are the closest to the experimental data, which can well describe the fatigue crack propagation process. The average relative error between the life curves obtained by different fitting methods and the experimental data curves is calculated, and the data are included in Table 8.

**Table 8.** The average relative error of the fatigue crack growth life curve simulated in different ways.

|  | 2 mm Reinforced Plate Structure | 3 mm Reinforced Plate Structure | 4 mm Reinforced Plate Structure |
|---|---|---|---|
| Linear fitting | 13.37% | 13.05% | 12.31% |
| Quadratic polynomial fitting | 11.13% | 7.66% | 10.76% |
| Cubic polynomial fitting | 8.04% | 3.33% | 5.69% |

As shown in Table 8, the relative error between the fatigue crack life curve calculated by the cubic polynomial fitting of all reinforced plate structures and the real test data is the smallest, which can meet the safety and accuracy requirements.

## 5. Conclusions

Through the crack propagation test of the LY12 aluminum alloy-reinforced plate structure, it was found that the fatigue crack propagation life and fatigue limit crack length of reinforced plate structures with different thicknesses will increase with the increase in the thickness of the reinforced plate. Under the same crack length, the fatigue crack growth rate decreases with the increase in the thickness of the reinforced plate.

The stress intensity factor amplitude $\Delta K$ of the reinforced plate structure with different thicknesses under different crack lengths can be calculated. As the crack increases, the stress intensity factor amplitude $\Delta K$ of the structure increases. Under the same crack length, the thicker the reinforced plate thickness, the smaller the stress intensity factor amplitude $\Delta K$. With the increase in crack length, the equivalent load at both ends of the central defect plate in the strengthened plate structure with different thicknesses is decreasing. Under the same crack length, the larger the thickness of the strengthened plate, the smaller the equivalent load of the central defect plate. With the expansion of the crack, the load ratio of the reinforcing plate with different thicknesses in the structure is increasing. Under the same crack length, the larger the thickness of the reinforcing plate, the higher the load ratio.

Using the secondary development program, the finite element analysis of the damage tolerance test of the LY12 reinforced plate structure is carried out. The stress intensity factor amplitude $\Delta K$ output by the program is compared with the $\Delta K$ calculated by the test data. The average relative errors of $\Delta K$ obtained by the two methods of the 2 mm, 3 mm, and

4 mm reinforced plate structure are 4.85%, 6.78%, and 4.57%, respectively, and the error is small.

Compared with the experimental data, the results of the finite element simulation are more conservative. The relative errors of the ultimate crack lengths of 2 mm, 3 mm, and 4 mm reinforced plate structures obtained by experiment and simulation are 5.33%, 7.42%, and 8.64%, respectively. The load distribution of the strengthened plate structure obtained by simulation and experiment is basically the same. The average relative errors of the equivalent loads on the two ends of the central defect plate in the 2 mm, 3 mm, and 4 mm reinforced plate structures under each crack length are 4.85%, 6.78%, and 4.57%, respectively.

Based on the cubic polynomial fitting formula, the fatigue crack growth life of the 2 mm, 3 mm, and 4 mm reinforced plate structure is estimated to be 37,041, 49,281, and 62,552, respectively. The relative errors between the fatigue crack growth life and the experimental fatigue crack growth life are 8.76%, 5.23%, and 1.28%, respectively. The obtained fatigue life is more conservative.

The average relative error between the fatigue crack growth life curve and the test data calculated by the function relationship between $a$ and $\Delta$ based on the cubic polynomial fitting of the reinforced plate structure with different thicknesses is the smallest, constituting 8.04%, 3.33%, and 5.69%, respectively.

The validity of the simulation method is verified by comparing the simulation results with the experimental results. It provides a method for the life prediction and crack propagation prediction of the damage repair method of strengthening plate structure.

**Author Contributions:** Methodology, F.W. and X.Z.; Software, C.L. and X.Z.; Validation, X.Z.; Formal analysis, C.L. and F.W.; Investigation, C.L. and S.Y.; Data curation, C.L.; Writing—original draft, C.L. and S.Y.; Writing—review and editing, F.W. and X.Z.; Supervision, F.W. and X.Z.; Project administration, X.Z. All authors have read and agreed to the published version of the manuscript.

**Funding:** This research was funded by the National Natural Science Foundation of China (Grant No. 11572253) and (Grant No. 11972302).

**Data Availability Statement:** The data that support the findings of this study are available from the corresponding author upon reasonable request.

**Acknowledgments:** The authors gratefully acknowledge the support sponsored by the National Natural Science Foundation of China (Grant No. 11572253) and (Grant No. 11972302).

**Conflicts of Interest:** The authors declare that they have no known competing financial interest or personal relationship that could have appeared to influence the work reported in this paper.

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
