# Peer review of "Studying a Repair Method of LY12 Aluminum Alloy Plate"

_metals, doi:10.3390/met13101758_

Round 1

Reviewer 1 Report

Dear authors, 

please check paper thoroughly for linguistical mistakes and some others (e.g. Mpa).

Other then that, I don't have complains.

English language is OK, in my opinion.

Author Response

I have learned much from editor’s and reviewers' comments, which are fair, encouraging and constructive. After carefully studying the comments and your letter, I have made corresponding corrections and further improved the work. Revised portion are marked in red in the revised manuscript. The main corrections in the manuscript and the point-to-point responds to reviewers' comments are as following:

-----Reviewer # 1

Dear authors,  please check paper thoroughly for linguistical mistakes and some others (e.g. Mpa).Other then that, I don't have complains.

Response:

Thank you for reviewer's constructive comment. I have corrected some unit symbols. and revised the whole manuscript carefully and tried to avoid any errors of grammar, syntax, or spelling. Now we believe the revised paper will provide a more readable description on the method and the main results of this study.

Reviewer 2 Report

The content of the manuscript not meeting the title. There is a lack of materials information in Materials and methods section.

Reviewer 3 Report

The submitted paper discusses about repair method of LY12 aluminum alloy plate. In my opinion, the paper has been fairly organized and can be accepted for publication. The following comments are suggested:

1)   The respected authors should bring some key results at the end of abstract section.

2)   The innovation of this paper should be clearly stated in the introduction section.

3)   The authors should Ref, numbers for the equations they used. For instance, Eqs. 1, 2, 3, …

4)   In page 3, line 115, the authors stated that “???? is the geometric correction coefficient” which is not correct.

5)   The boundary conditions for FEM analysis should be clearly stated in the text.

6)   In Fig. 4, the authors should explain why the fatigue performance of the specimens increases with the increase of the thickness of the reinforced plate. Is there any critical thickness of the reinforced plate for the specimens?

7)   In Fig. 5, Under the same crack length, why is the smaller the stress intensity factor amplitude â–³K related to the thicker reinforced plate?

8)   The authors should explain why the difference in equivalent stress at both ends at the central defect plate between simulation and experiment is increased with the increase of crack length (please see figures 15, 16, 17).

9)     My overall comment is that the discussion of the provided figures is superficial. It should be a deep discussion on obtained data.

Round 2

Reviewer 2 Report

accept

Reviewer 3 Report

I am OK with the revisions. The paper can be considered for publication in the current format.